# Gram-Negative Bacterial Envelope Homeostasis under Oxidative and Nitrosative Stress

**DOI:** 10.3390/microorganisms10050924

**Published:** 2022-04-28

**Authors:** Thibault Chautrand, Djouhar Souak, Sylvie Chevalier, Cécile Duclairoir-Poc

**Affiliations:** Research Unit Bacterial Communication and Anti-infectious Strategies (UR CBSA), Rouen Normandy University, Normandy University, 55 rue Saint-Germain, 27000 Evreux, France; thibault.chautrand1@univ-rouen.fr (T.C.); djouhar.souak@univ-rouen.fr (D.S.); sylvie.chevalier@univ-rouen.fr (S.C.)

**Keywords:** Gram-negative bacteria, bacterial envelope, ROS, RNS, oxidative stress, nitrosative stress

## Abstract

Bacteria are frequently exposed to endogenous and exogenous reactive oxygen and nitrogen species which can damage various biomolecules such as DNA, lipids, and proteins. High concentrations of these molecules can induce oxidative and nitrosative stresses in the cell. Reactive oxygen and nitrogen species are notably used as a tool by prokaryotes and eukaryotes to eradicate concurrent species or to protect themselves against pathogens. The main example is mammalian macrophages that liberate high quantities of reactive species to kill internalized bacterial pathogens. As a result, resistance to these stresses is determinant for the survival of bacteria, both in the environment and in a host. The first bacterial component in contact with exogenous molecules is the envelope. In Gram-negative bacteria, this envelope is composed of two membranes and a layer of peptidoglycan lodged between them. Several mechanisms protecting against oxidative and nitrosative stresses are present in the envelope, highlighting the importance for the cell to deal with reactive species in this compartment. This review aims to provide a comprehensive view of the challenges posed by oxidative and nitrosative stresses to the Gram-negative bacterial envelope and the mechanisms put in place in this compartment to prevent and repair the damages they can cause.

## 1. Oxidative and Nitrosative Stresses: An Omnipresent Challenge

Oxidative stress is the result of an imbalance between the presence of reactive oxygen species (ROS) and the ability of a biological system to detoxify them and their side products. Similarly, nitrosative stress is an imbalance of reactive nitrogen species (RNS). Some of these molecules can play an important role in signalization in both eukaryotes and prokaryotes. For example, nitric oxide (NO) is involved in the dispersion of biofilm in multiple bacterial species [1] and in the regulation of vasoconstriction in mammals [2,3]. However, the strong redox potential of ROS and RNS allows them to damage a wide range of biomolecules, and any excess can jeopardize cell homeostasis. As a result, most biological organisms possess defense mechanisms allowing them to manage these stresses. Some reactive oxygen and nitrogen species are endogenously produced by the bacterial metabolism, in particular in aerobic bacteria. In addition, bacteria can often encounter exogenous oxidative and nitrosative stresses in their environment, one glaring example being cellular immune response.

The envelope is a complex structure whose integrity is critical for cell survival, serving as a selective barrier at the interface between the inner and the outer bacterial environments. It is also the first bacterial structure to encounter ROS and RNS. Correct membrane function is essential for the survival of bacteria, as well as peptidoglycan integrity and synthesis. Since oxidative and nitrosative stresses can affect a large variety, if not all of the envelope components, it is critical for the cell to maintain envelope homeostasis to survive these stresses. This review aims at gathering the current understanding of the impacts of oxidative and nitrosative stresses on the bacterial envelope and the bacterial mechanisms implemented to cope with such stresses. This review first will address the main actors of oxidative and nitrosative stresses and their exogenous sources, then it will focus on the damages induced by RNS and ROS on the envelope of Gram-negative bacteria, including numerous challenging bacteria in terms of health, and finally, it will address the bacterial response mechanisms to maintain envelope homeostasis, whether by preventing or repairing these damages.

## 2. The Gram-Negative Bacterial Envelope

The envelope of Gram-negative bacteria is composed of the outer (OM) and inner (IM) membranes and a thin peptidoglycan (PG) layer located in the periplasmic space, which is a highly oxidizing compartment.

### 2.1. The Outer Membrane (OM)

The OM is asymmetric, with an inner leaflet made of phospholipids facing the periplasm, and an outer layer of anionic glycolipids, including lipopolysaccharides (LPS). Typically, LPS are composed of a lipid A, also known as endotoxin, a core oligosaccharide, and a polysaccharide forming a distal chain known as O-antigen [4]. The OM functions as a selective physical permeability barrier, by controlling the entry into the cell. Most OM transmembrane proteins are β-barrel proteins, which are composed of β-sheets forming a cylinder [5]. *Escherichia coli* possesses approximately 100 of these proteins, whose functions are mostly unknown. The known β-barrel proteins notably include three major porins: OmpF and OmpC, which both allow the selective passage of small cationic molecules, and PhoE, which is dedicated to the entry of anionic molecules [6]. The other main group of proteins present on the OM are lipoproteins, which possess lipid moieties attached to a terminal cysteine residue [7]. Lipoproteins were considered to be overwhelmingly bound to the IM, and it is only recently that the importance of OM lipoproteins has been highlighted. Lipoproteins are anchored to the OM inner leaflet by their lipid tail facing the periplasm for most of them, while others, called surface lipoproteins, decorate the outer surface of Gram-negative bacteria [8]. They are involved in various processes such as nutrient acquisition, cell signaling, and virulence [8,9]. These functions are not exclusive to protein, however, as phospholipids also participate in the homeostasis of both the outer and inner membranes [10].

### 2.2. The Inner Membrane (IM)

The IM is composed of a symmetrical bilayer of phospholipids and possesses a much greater protein diversity than the OM. As bacteria lack the intracellular organelles present in eukaryotic cells, the membrane-associated functions of bacterial metabolism take place at the IM [11]. These functions notably include lipid biosynthesis, energy production, and protein secretion. All the components of the Gram-negative bacteria envelopes are synthetized directly in the cytoplasm or at the inner leaflet of the IM. Therefore, all the envelope components must be translocated, secreted, or flipped through the IM. Furthermore, the IM acts as a second layer of protection and selection against the entry of exogenous molecules [12].

### 2.3. The Periplasm and the Peptidoglycan (PG)

Located between both membranes, the periplasm is an oxidative environment containing a high density of proteins. In this environment, the structure of proteins containing cysteines is usually stabilized by disulfide bonds. The periplasmic compartment fills various roles for the cell. Notably, it sequesters enzymes that would be toxic to the cell in the cytoplasmic space, such as RNAses and alkaline phosphatases, and allows specific protein folding and oxidation and regulation of cell division [13]. In addition, the periplasm harbors the PG, which allows the cell to retain its shape. PG is composed of polymerized glycans that form linear strands cross-linked by short peptides. These glycans strands are made of β-1,4-connected N-acetylglucosamine residues alternating with N-acetylmuramic acid residues [12]. PG thus forms a net-like polymer around the cell that protects it against turgor and serves as a scaffold to attach proteins, lipoproteins, and other molecules [12]. Its basic composition is well conserved between bacteria, but specific variations can alter its properties.

The envelope serves as an interface between the cell and the environment by selecting which substances are able to cross it. Furthermore, the envelope is the first cell component to be affected by the cell environment and must be able to withstand the various stresses it encounters via complex regulatory pathways known as the cell envelope stress response (ESR) [14]. As a result, maintaining membrane homeostasis is critical for the cell. Aerobic bacteria are commonly exposed to oxidative and nitrosative stresses from their environment, resulting in the excessive presence of reactive oxygen or nitrogen species, respectively. These processes can alter multiple envelope components, such as phospholipids and proteins, causing damage to this cell compartment [15,16]. It is necessary to consider the complex interactions between these species and biological molecules to understand the challenges bacteria face.

## 3. Reactive Oxygen and Nitrogen Species: A Complex Reactions Network

### 3.1. Oxidative Stress Chemistry

Molecular oxygen (O_2_) is a small non-polar molecule that diffuses freely across usual biological membranes [17]. Therefore, the bacterial intracellular O_2_ concentration is similar to their environment. As a result, bacteria either have to avoid oxidative stress by living in anaerobic or microaerobic environments or survive elevated internal oxygen levels. Oxygen is not toxic by itself, as it is practically unreactive with the molecules structuring biological organisms that are lipids, proteins, carbohydrates, and nucleic acids. However, the reduction of O_2_ can generate various ROS, such as the superoxide anion (O_2_^•^^−^), hydrogen peroxide (H_2_O_2_), and hydroxyl radicals (OH^•^) [18]. In aerobic environments, endogenous ROS may be produced in bacteria via the reaction between O_2_ and univalent electron donors. These donors can be metal centers, dihydroflavin cofactors, or quinones [18]. The main endogenous source of O_2_^•^^−^ and H_2_O_2_ is the autoxidation of non-respiratory flavoproteins [19] by electron transfer between O_2_ and the dihydroflavin of the reduced flavoproteins. This reaction leads to the formation of O_2_^•^^−^, which generally goes through another electron transfer before escaping the active site of the enzyme, generating H_2_O_2_. Aside from its conversion into H_2_O_2_, O_2_^•^^−^ is in equilibrium with the hydroperoxyl radical (HO_2_^•^) through the reversible reaction O_2_^•−^ + H^+^ ⇋ HO_2_^•^ [20]. Since the pKa of this reaction is 4.88, it is estimated that the HO_2_^•^ form represents less than 1% of the total superoxide in the cellular cytoplasm [21]. However, cytoplasmic pH may not be uniform and may be significantly lowered in the proximity of membranes containing negatively charged phospholipids such as cardiolipin, phosphatidylserine, and phosphatidylinositol [22]. Therefore, local pH decrease could allow the formation of significant quantities of HO_2_^•^ near the IM. Unlike O_2_^•−^, HO_2_^•^ is hydrophobic, allowing it to cross to the IM lipidic core. As a result, HO_2_^•^ is critical in the process of lipid peroxidation. In the presence of light, photosynthetic organisms also generate the highly reactive singlet oxygen (^1^O_2_) through the pigments of their photosystems [23]. This stress can also occur in non-photosynthetic microorganisms through other cellular cofactors such as rhodopsin, quinones, flavins, and porphyrins [23].

### 3.2. Nitrosative Stress Chemistry

The range of reactive molecules created by oxidative stress is not limited to ROS. Indeed, RNS production is tightly linked to the presence of ROS in the cells. Nitric oxide (NO) is a small lipophilic radical which diffuses across biological membranes. It is an important molecule for signalization in biological organisms. In mammals, NO notably controls blood pressure and acts as a messenger in the central nervous system [24]. Despite its role as a signaling molecule, NO is toxic for biological organisms at high concentrations, and this molecule is synthetized by macrophages to combat pathogens during the immune response through their inducible nitric oxide synthases (iNOS), making resistance to RNS critical for pathogens [25,26]. NO is more reactive than oxygen regarding the structural components of biological organisms, particularly proteins. Its toxicity comes from its ability to inhibit haem enzymes binding dioxygen, react with Fe-S centers, and indirectly induce the nitrosation of proteins [27]. Similar to oxygen, NO is relatively unreactive with most biological molecules. Direct biological targets of NO are limited to radicals and metal complexes, especially Fe-containing complexes. However, these reactions with intracellular molecules can generate other reactive species much more harmful to the cell. For example, NO with O_2_^•−^ reacts at a diffusion-controlled rate to form the peroxynitrite anion (ONOO^−^) and its conjugated acid, peroxynitrous acid (ONOOH) (Figure 1) [28]. Since these two species coexist and are in an acid-base equilibrium in common biological conditions, the term peroxynitrite is generally used to describe both species [16]. Peroxynitrite is an impactful biological oxidant. ONOOH isomerizes to nitrate at rates of 0.095 s^−1^, 1.3 s^−1^, and 4.5 s^−1^ at 5 °C, 25 °C, and 37 °C, respectively [29] through the formation of both OH^•^ and NO_2_^•^, hereafter simply noted OH and NO_2_ [16]. However, in biological conditions, peroxynitrite reacts with its biological targets, such as proteins, lipids, and nucleic acids, at a much faster rate, making this reaction irrelevant. Due to its high reactivity, peroxynitrite-induced damages will mostly depend on the kinetics of the reactions between peroxynitrite and its surrounding targets in priority protein metal centers, thiols, and selenols [16]. Another important RNS is NO_2_. This may appear alongside OH, in an aqueous environment, through the isomerization of ONOOH (Figure 1). In an air polluted by NO, such as smog, for instance, NO autoxidation in the gaseous phase leads to the formation of NO_2_ through the equation 2NO+O_2_ → 2NO_2_. Further reactions between NO and NO_2_ leads to the formation of N_2_O_3_ through the equation NO_2_+NO ⇋ N_2_O_3_ [30]. In aqueous media, such reactions may occur too fast to produce a significant amount of NO_2_ intermediate. However, these reactions still have biological relevancy as they may take place in non-aqueous media such as the lipid bilayers of membranes [31].

### 3.3. Exogenous Sources of Oxidative/Nitrosative Stress

#### 3.3.1. Biotic Stress Sources

In addition to the endogenous ROS and RNS synthesis through metabolic processes described above, bacteria have to deal with exogenous stresses from various sources.

The most studied one is the oxidative burst released by phagocytic cells during the immune response. During this process, macrophages and neutrophils cause an oxidative burst on phagocyted bacteria while they synthetize O_2_^•−^ through the activity of the membrane nicotinamide adenine dinucleotide (NADPH)-oxidase. Half of the O_2_^•−^ reacts with H^+^ to form H_2_O_2_. In addition to ROS, macrophages also release large quantities of NO, synthetized by their iNOS. The simultaneous release of O_2_^•−^ and NO in biological conditions lead to the formation of peroxynitrite [16]. The mammalian immune system also induces oxidative stress inside bacteria using PG Recognition Proteins (PGRPs). These proteins kill bacteria by induction of oxidative, thiol, and metal stresses in bacteria. PGRPs induce oxidative stress by blocking the bacterial respiratory chain, which promotes the production of H_2_O_2_ inside the cell [32]. In the context of host infection, bacteria are also exposed to other envelope-damaging factors such as the membrane attack complex and lysozyme, which synergize to degrade bacterial envelopes [33]. In addition to its role in bacterial killing through peptidoglycan hydrolysis, lysozyme possesses immune-dampening properties. It is notably able to decrease the neutrophil oxidative burst [34] and neutralize the prooxidant advanced glycation end products [35], which block its bactericidal activity [36].

The use of ROS as a weapon against microorganisms is not exclusive to the animal immune system. Some plants and microorganisms excrete ROS or redox-cycling compounds to suppress the growth of their competitors. For example, Gram-positive lactic acid bacteria are able to release important concentrations of H_2_O_2_ in their environment. In fact, they lack full respiratory chains, but many of them still use oxygen as a direct electron acceptor, thanks to lactate and pyruvate oxidases [37,38]. The obtained product is converted by an acetate kinase in another ATP molecule(s), as well as H_2_O_2_. In favorable experimental conditions, lactic acid bacteria can produce millimolar concentrations of H_2_O_2,_ inhibiting the growth of other bacteria [37,39,40]. However, in natural conditions, H_2_O_2_ might be carried away or otherwise degraded; therefore, such concentrations might not be reached. Another example is the production of pyocyanin by *P. aeruginosa*. This secondary metabolite is able to inhibit the respiration chain, leading to the impairment of energy-dependent transport systems and the production of oxidative species, such as H_2_O_2_ [41,42].

Another source of oxidative stress could be antibiotics. Oxidative stress is not directly the primary mode of action of antibiotics to suppress bacterial growth, although new antimicrobial relying on the production of ROS and RNS are gaining interest [43]. Indeed, antibiotics mainly target peptidoglycan biosynthesis, protein synthesis, or DNA replication and repair. However, oxidative stress could be a secondary effect of some antibiotics. For instance, some bactericidal antibiotics [44,45] could lead to the intracellular production of ROS such as O_2_^•−^ and H_2_O_2_ [43,46,47], although these results have been challenged [44,45]. Furthermore, the response to some antibiotics depends on their species [48]. Overall, the impact of oxidative stress in the modes of action of traditional antibiotics on Gram-negative bacteria is a complex question requiring further clarification. In addition to their oxidative potential, antibiotics often induce protective responses leading to the reshaping of the envelope, especially by modulating the efflux pumps and porins at the outer membrane through the triggering of envelope regulators, such as the Cpx complex [49].

#### 3.3.2. Abiotic Stress Sources

Environmental bacteria are also subjected to oxidative and nitrosative stresses through multiple abiotic sources. An important source is ultraviolet (UV) radiations, which can induce oxidative stress inside the cells by direct exposure or through the generation of H_2_O_2_ in surface water through UV photochemistry [50,51,52]. Anthropogenic activities also create the conditions for oxidative and nitrosative stresses in the environment. The most glaring example is the release of vast amounts of ROS and RNS into the atmosphere by fuel combustion processes used in transport and industry [53,54], which can affect bacterial physiology [55,56].

### 3.4. Targets of ROS and RNS

Overall, the exposure of bacteria to excessive oxidative and nitrosative stresses destabilize the envelope, impairing its proper functions, inducing membrane permeabilization [56] and hyperpolarization [57] and eventually leading to cell death. The harm caused by ROS and RNS to the bacterial envelope comes from their ability to react with biomolecules and alter their biochemical properties, disturbing the biochemical processes necessary for cell survival. The vast range of ROS and RNS leads to a wide panel of biological targets. In the envelope, the main targets for oxidative and nitrosative stresses are proteins, and in some cases, phospholipids. These molecules can undergo various alterations discussed below, constituting a real challenge for cells to maintain their envelope integrity.

#### 3.4.1. Phospholipids

ROS and RNS are relatively unreactive with the phospholipids forming biological membranes, except for phospholipids containing polyunsaturated fatty acids (PUFAs), on which ROS and RNS induce their lipid peroxidation. Most bacteria do not synthetize PUFAs, and as a result, most bacterial membranes are composed of saturated or monounsaturated phospholipids [58], thence theoretically insensitive to ROS and RNS. However, studies on *P. fluorescens* showed modifications of the membrane phospholipid composition after exposure to NO_2_, despite the absence of PUFAs in the strain [55,56]. By contrast, in various water sources and fish microbiota, bacteria, mostly from the *Shewanella* genus, synthetize PUFAs, such as eicosapentaenoic acid and docosahexaenoic acid, which are prone to peroxidation [59]. Moreover, even bacteria that are unable to synthetize them may incorporate PUFAs from their environment into their membranes. For example, various *Vibrio* species, such as *V. cholerae*, *V. vulnificus*, and *V. parahaemolyticus*, possess the mechanisms required to accumulate and incorporate PUFAs into their membranes [60].

Noticeably, the vertebrate immune system takes advantage of such a PUFA incorporation mechanism to initiate lipid peroxidation in bacteria. In vertebrates, arachidonic acid is released concomitantly with RNS and ROS during the oxidative burst. In the context of the immune response, PUFAs are toxic for a wide range of bacteria such as *Acinetobacter baumannii* [61], *Listeria monocytogenes* [62], *Pseudomonas aeruginosa* [62], including gram-positive species, such as *Staphylococcus aureus* [63], *Cutibacterium acnes* [63], and *Streptococcus pneumoniae* [64]. The PUFAs toxicity depends on their capacity to esterify the fatty acids into the bacterial membranes [65]. In the case of the Gram-positive *Staphylococcus aureus*, the toxicity of PUFAs was shown to be specifically mediated by lipid peroxidation [66]. As a result, even though bacteria are relatively less sensitive to lipid peroxidation than other genera, this process still plays a crucial role in bacterial pathogenicity.

#### 3.4.2. Peptidoglycan

Although the effects of oxidative and nitrosative stress on peptidoglycan are poorly understood, a few studies reported links between oxidative or nitrosative stress and proteins associated with peptidoglycan or its synthesis [67,68,69,70]. The clearest demonstrated effect of oxidative stress on peptidoglycan comes from a recent study by Giacomucci et al. This study showed that the absence of the protein ElyC of unknown function induces the overproduction of OH^•^ in the periplasm, which leads to a direct or indirect interruption of peptidoglycan synthesis [71].

#### 3.4.3. Envelope Proteins

ROS and RNS can react with a wide variety of protein features and virtually with any amino acid. However, the susceptibility of amino acids to ROS and RNS oxidation varies, and the reactions taking place are determined by their reaction rate. ROS react especially well with the amino acids containing sulfur, methionine, and cysteine [72]. The oxidation of methionine residues by ROS forms one of two diastereoisomers of methionine sulfoxide (Met-O): methionine-S-sulfoxide or methionine-R-sulfoxide. None of the couple of Met-O stereoisomers is preferably formed over the other, but different reductases are required to reduce each one [73]. ROS oxidation of cysteine residues first generates a highly reactive sulfenic acid derivative (RSOH). Depending on its microenvironment, RSOH may also further react with a nearby cysteine to form a disulfide bond or be oxidized to sulfinic (RSO_2_H) and sulfonic (RSO_3_H) acid. Sulfenic acid and methionine sulfoxide formation may induce misfolding, inactivation, or degradation of proteins [74,75].

#### 3.4.4. Protein Carbonylation

ROS can also induce irreversible protein modifications through protein carbonylation. So ROS react with the side chains of amino acids to form carbonyl groups [76]. Protein carbonylation is directly induced by ROS such as H_2_O_2_ on the side chains of amino acids: arginine, lysine, proline, and threonine [77]. Furthermore, aldehydes formed by lipid peroxidation can also indirectly target the side chains of lysine, histidine, and cysteine through the Michael reaction [78,79]. Similar to methionine sulfoxide and sulfenic acid, carbonylation can impair protein functions. Protein carbonyls are more difficult to induce than methionine sulfoxide and sulfenic acid and then are considered markers of a more intense oxidative stress [80]. In *E. coli*, for example, the entry into the stationary phase caused by nitrogen or carbon starvation leads to an increase in protein carbonylation [81].

#### 3.4.5. Protein S-Nitrosylation

Cysteines are also prone to S-nitrosylation, a process by which NO moieties covalently bind to thiols to form an S-nitrosothiol (SNO). Its formation is related to reaction with different RNS, such as NO and N_2_O_3_, or NO carriers such as other nitrosothiols. This process may alter protein function. Protein S-nitrosylation is reversible and is often enzymatically mediated by organisms to protect their proteins against untargeted S-nitrosylation from exogenous nitrosative stress [82].

#### 3.4.6. Tyrosine Nitration

The most characteristic protein modification induced by peroxynitrite is tyrosine nitration. Its first step consists of the one-electron oxidation of the tyrosine phenolic ring to form a tyrosine radical, Tyr^•^. This step is performed by various ROS and RNS, such as OH, NO_2_, or secondary produces of peroxynitrite reactions, CO_3_^−^_,_ and oxo-metal complexes. In hydrophobic environments, tyrosine nitration may also be initiated by the intermediates of lipid peroxidation, such as lipid peroxyl LOO^•^ and alkoxyl LO^•^ [83,84]. These radicals contribute to tyrosine nitration within lipid bilayers. During the second step of the tyrosine nitration, Tyr^•^ reacts with NO_2_ to form NO_2_Tyr. The complete process impairs protein function and leads to loss of function but is compensated by a gain in proteins implicated in regulatory cascades [16].

## 4. Defenses against Oxidative and Nitrosative Stresses

Considering the wide array of alterations that the envelope biomolecules can undergo under oxidative and nitrosative stresses, bacteria have developed numerous mechanisms to protect themselves and maintain their envelope integrity. These mechanisms rely on two main strategies. The first strategy consists of the scavenging and reduction of ROS and RNS to prevent the biomolecules damages induced by these species with peroxidases and superoxide dismutases. The second strategy consists of the repair of damaged biomolecules. These mechanisms are under the regulation of global oxidative stress sensors and regulators such as OxyR, SoxRS, or RpoS, while others are regulated by the ESR pathways [15,85].

### 4.1. Reducing Systems

#### 4.1.1. Superoxide Dismutases

The first defense of bacteria against ROS and RNS is scavenging enzymes: superoxide dismutases (SODs). They are synthetized by most organisms to catalyse the dismutation of O_2_^•−^ into O_2_ and H_2_O_2_ (Figure 2). Since O_2_^•−^ does not diffuse through biological membranes easily, different SODs are present in different cell compartments [86]. In *E. coli*, SodA and SodB are cytoplasmic [87,88], while SodC is targeted to the periplasm [89]. In physiological conditions, SodC is assumed to reduce O_2_^•−^ released in the cytoplasm by the respiratory chain. Periplasmic SODs are involved in the resistance of *Salmonella typhimurium to macrophages*. SodC mutants are especially susceptible to the combination of O_2_^•−^ and NO, suggesting that SodC protects the cell against peroxynitrite formation by diverting O_2_^•−^ [90].

#### 4.1.2. Catalases/Peroxidases

Most organisms possess peroxidases and catalases able to reduce H_2_O_2_. *E. coli* possesses several cytoplasmic peroxidases such as (i) alkyl hydroperoxide reductase AhpCF, which reduces H_2_O_2_ into water, (ii) thiol peroxidase Tpx, which is involved in the reduction of bulky hydroperoxides, and (iii) peroxiredoxin Bcp which reduces a broad range of molecules with lesser efficiency [15]. While cytoplasmic peroxidases are common, few have been found in the periplasm [15]. This could be explained by the ability of H_2_O_2_ to easily cross biological membranes, making the compartmentalization of peroxidases less relevant to the promotion of damage. However, the discovery of periplasmic peroxidases indicates that some bacteria require H_2_O_2_ to be scavenged in this compartment. The first protein identified as a periplasmic peroxidase was PprX, which reduces H_2_O_2_ and cumene hydroperoxide [91].

#### 4.1.3. NO-Reductases

Most bacteria are able to use nitrogen anions as final electron acceptors in the respiratory chain under limited O_2_ conditions [92]. This process, named denitrification, involves successive reductions from NO_3_^−^ to NO_2_^−^, NO, N_2_O, and finally N_2_. Denitrifying bacteria possess nitric oxide reductases (Nor) divided into two families depending on the use of electron donors. In fact, qNor uses quinol, whereas cNor uses soluble proteins such as cytochrome c [93]. While Nor predominantly acts in the respiratory pathway, it plays a role in resistance to endogenous and exogenous nitrosative stress. This resistance proves useful in bacterial pathogenesis, as Nor is linked to the virulence of several bacteria, such as *E.coli* and *P. aeruginosa* [94].

#### 4.1.4. Periplasmic Cytochrome c Nitrite Reductase

The periplasmic cytochrome c nitrite reductase NrfA can convert nitrite (NO_2_^−^) enzymatically into ammonium (NH_4_^+^). Unlike the nitrite reductase, Nir presents in denitrifiers and converts nitrites to NO; the reaction mechanism of NrfA is thought to occur in multiple steps of electron transfer from its five hemes to its substrate. This reaction creates NO and hydroxylamine (NH_2_OH) intermediates. It has also been shown that NrfA is able to convert NO into NH_4_^+^. The ability of NrfA to detoxify NO at the periplasmic level is important for the growth of *E. coli* in high NO concentrations [95]. This protein also plays an important role in the resistance of *Campylobacter jejuni*, *Salmonella Enterica*, and *Wolinella succinogenes* to nitrosative stress [96,97,98]. Despite the ability of NO to cross bacterial membranes, localized detoxification remains crucial for cell survival. An explanation could be that some NO-derived reactive species, such as ONOO^−^, are not membrane permeant.

#### 4.1.5. Cytochrome *bd*

Cytochrome *bd* is a tri-haem membrane protein and is a terminal quinol oxidase of the respiratory chain in bacteria under poor growth conditions. This enzyme reduces molecular oxygen to water by using quinols as electron donors [99]. Cytochrome *bd* possesses a peroxidase activity, decomposes both H_2_O_2_ and ONOO^−^, and converts NO into NO_2_^−^ (Figure 2) [100]. Therefore, cytochrome *bd* is involved in the virulence and resistance to the immune response of various bacterial species, such as *Shigella flexneri* [101], *Brucella abortus* [102], or *Salmonella enterica* [103].

### 4.2. Repair Systems

#### 4.2.1. Protein Repair Mechanisms

Unrepaired sulfonic acid, Met-O, S-nitrosylation, and tyrosine nitration may lead to protein misfolding, inactivation, or degradation [74,75]. The periplasm is a more oxidizing environment than the cytoplasm [104], and most cysteine residues of the periplasmic proteins are engaged in disulfide bonds, preventing their oxidation by ROS. However, a small proportion of periplasmic proteins possess cysteine residues in their reduced thiol form [105], leaving them vulnerable to oxidation by ROS. Periplasmic proteins depending on a cysteine residue for their activity, are protected by a system regulating their redox state. Disulfide bond formation protein A (DsbA) possesses a labile disulfide bond transferred to proteins that translocate to the periplasm. The reduced DsbA is then recycled by the IM DsbB. DsbA preferentially introduces disulfide bonds between consecutive cysteine residues, resulting in potential mismatches [15]. Such mismatches are then corrected by the chaperone disulfide isomerase DsbC, which also repairs oxidized cysteine residues. One of the few periplasmic proteins possessing reduced thiols is YbiS, an L,D-transpeptidase catalyzing the covalent attachment between the lipoprotein Lpp and peptidoglycan discovered in *E. coli*. The catalytic activity of YbiS requires only the cysteine residue of this enzyme [106].

Met-O residues formed by ROS and RNS are reduced to methionine by the methionine sulfoxide reductase (Msr) system. This system revolves around the activity of MsrA and MsrB polypeptides, two methionine sulfoxide reductases with no identity at the sequence or structural level [107]. MsrA specifically reduces methionine-S-sulfoxide while MsrB specifically reduces methionine-R-sulfoxide (Figure 3). Both polypeptides are highly conserved between organisms; however, the genetic organization of *msr* varies widely among bacteria [108]. Some bacteria, such as *E. coli*, transcribe *msrA* and *msrB* independently, whereas others, such as *Helicobacter pylori* and *Treponema pallidum*, *msrA* and *msrB* compose a single transcription unit, resulting in the synthesis of a polypeptide with two active sites. The localization of active Msr polypeptides is also variable depending on the species. In *H. pylori*, Msr possesses SecA-dependent signal sequences involved in the secretion of Msr. The polypeptide is present in the cytoplasmic and membrane fractions but is only active in the membrane fraction [109,110]. In contrast, no activity was found in the membrane fraction of *Actinobacillus actinomycetemcomitans*, while the cytoplasmic and periplasmic fractions showed Msr activity [111]. Furthermore, Msr was also detected on the surface of bacteria for the first time on *Ochrobactrum anthroporium* [112].

#### 4.2.2. PG Repair Mechanisms

Aconitases (Acn) are iron-sulfur proteins present in eukaryotes and prokaryotes. They are responsible for the conversion of citrate into isocitrate and, inversely, during the Krebs cycle [113]. When their [4FeS] cluster is lost, some Acns are also able to bind to specific mRNA sequences [114]. The best-studied Acn is the eukaryotic iron-responsive protein 1 (IRP1) [113]. The formation of apo-Acns occurs during exposure to ROS or RNS, and after long-term iron starvation [115]. In *H. pylori*, AcnB regulates the expression of the PG deacetylase PgdA under oxidative conditions [116]. PgdA is involved in the preservation of peptidoglycan integrity through the deacetylation of peptidoglycan N-acetylglucosamine residues [117]. This deacetylation prevents the formation of β-1,4 bonds between N-acetylglucosamine and N-acetylmuramic acid residues, which are the target of hydrolysis by lysozyme [117]. The deacetylation catalyzed by PdgA also mitigates host immune detection and increases the survival of *H. pylori* in murine stomachs [118].

#### 4.2.3. Membrane Repair Mechanisms

In *E.*
*coli*, the NADH peroxidase Ahp catalyzes the reduction of H_2_O_2_ to water using NADH, which is converted into unstable NAD^+^ [119]. In addition to its role in limiting the amount of ROS in the cell, Ahp also plays an important role in the protection of membranes against oxidative stress by reducing fatty acid peroxides (Figure 3) [120]. 

### 4.3. Envelope Stress Response (ESR)

Cellular homeostasis is dependent on the integrity of the cell envelope. Bacteria possess a wide array of mechanisms to cope with various stresses that can compromise their envelope integrity [121]. The various pathways involved in the maintenance of the envelope under stress form the ESR [14,122]. Outside of stressful environments, ESRs also play a housekeeping role by regulating envelope precursors and ensuring adequate envelope biogenesis. These pathways possess specificities but often overlap and are not necessarily specific to a single stress. Several of these ESRs regulate proteins involved in protection against oxidative-stress damage [85].

#### 4.3.1. Cpx Complex in *E. coli*

The Cpx ESR is controlled by a two-component system involving CpxA and CpxR. CpxA is a histidine kinase sensor bound to the IM, and CpxR is its cytoplasmic DNA-binding response regulator [123]. Briefly, when activated, CpxA autophosphorylates conserved histidine residues, and the resulting phosphate is transferred to a conserved aspartate residue of CpxR. The phosphorylated CpxR then regulates the transcription of over 100 genes to relieve envelope stress. The Cpx response also impacts respiratory complexes in *E. coli* [124]. Notably, CpxRA causes the mitigation of PGRP-induced oxidative stress, which blocks respiratory chains [32].

#### 4.3.2. σ Factors

Bacterial sigma factors are transcription factors that regulate a set of genes in response to specific stimuli, including cell-envelope stress responses. They are subunits of the RNA polymerase, allowing them to target specific promoters. Sigma factors are divided into the σ^70^ and σ^54^ families based on their similarities with those of *E. coli* in terms of structure, amino acids sequence, and mechanism of action. The σ^70^ family is split into four phylogenetic groups. Group 4, also called the extracytoplasmic function (ECF) sigma factors subfamily, is the largest of these groups and includes the more divergent sigma factors that are involved in various environmental responses. ECFs are extremely diverse, and a lot of them are still unknown from a mechanistic and functional point of view. ECFs take part in a wide range of processes, including starvation response, metal homeostasis, and virulence [14]. ECF activity is regulated by sequestration with a specific inhibitor called the anti-sigma factor, which is often membrane-bound and co-transcribed with its relative ECFs.

The most studied ECF RpoE, also called σ^E^, is involved in the protection of the cell against various stresses. RpoE was originally discovered in *E. coli* for its ability to induce the transcription of the gene of the sigma factor RpoH, responsible for the transcription of heat-shock genes, at lethal temperatures (50 °C) [125]. Today, the best-characterized model of activation for RpoE is through the proteolysis of its anti-sigma factor RseA by both DegS and RseB. These two proteins are activated by misfolding of OM proteins and by the accumulation of LPS in the periplasmic compartment, respectively [126]. Overall, RpoE detects signals caused by OM malfunction and activates damage repair pathways. RpoE notably elicits an envelope stress response in response to the misfolding of OM proteins induced by oxidative stress. In the photosynthetic bacterium *Rhodobacter sphaeroides*, RpoH activates oxidative stress defenses under the control of RpoE in response to ^1^O_2_ [127]. RpoE also regulates the expression of the highly conserved chaperone protease, DepG. This periplasmic protein is found in most Gram-negative bacteria and ensures survival in animal hosts. The function of DepG is to degrade misfolded proteins in the periplasm to prevent their accumulation.

Other sigma factors respond to oxidative stress. Notably, oxidative stress activates the ECF σ^T^, which promotes the transcription of various genes, including those involved in membrane homeostasis [128]. Furthermore, the ECF RpoS regulates the C-tail anchored IM protein ElaB in coordination with OxyR. While its precise function is unclear, ElaB is upregulated during oxidative stress and seems to play a role in the preservation of membrane integrity under oxidative stress, and is strongly upregulated in ampicillin-tolerant *E. coli* [129,130].

## 5. Conclusions

Exogenous ROS and RNS are a frequent occurrence in bacterial life, whether they come from (i) the immune system of their host, (ii) other microorganisms, or (iii) abiotic sources. The oxidative and nitrosative stresses induced by such molecules have a wide array of deleterious effects for bacteria, caused by a complex network of reactions both between these reactive species and bacterial biomolecules. ROS and RNS can notably alter the proteins and lipids composing the Gram-negative bacterial envelope. Maintaining the integrity of the envelope is crucial for these bacteria to survive under oxidative and nitrosative stresses, and a wide array of protection mechanisms have been selected by evolution to lessen and repair the damages caused by these stresses. Overall, the actors of both oxidative and nitrosative stresses resistance in the cytoplasm are better known than those acting at the level of the periplasm. However, more and more proteins involved in oxidative and nitrosative stresses resistance in the periplasm are discovered, often with functions mimicking their cytoplasmic counterparts. This suggests that the localization of the oxidative and nitrosative stresses response is a crucial factor for cell survival, even when dealing with reactive species able to pass through biological membranes.

## Figures and Tables

**Figure 1 microorganisms-10-00924-f001:**
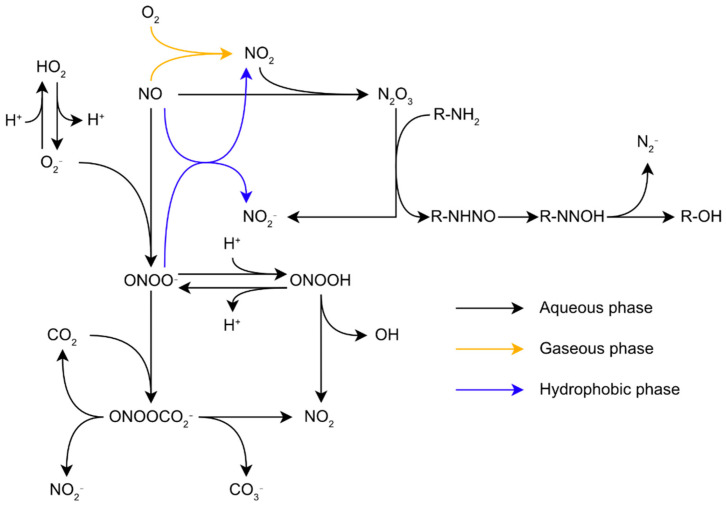
Reactions network of reactive oxygen and nitrogen species in biological conditions.

**Figure 2 microorganisms-10-00924-f002:**
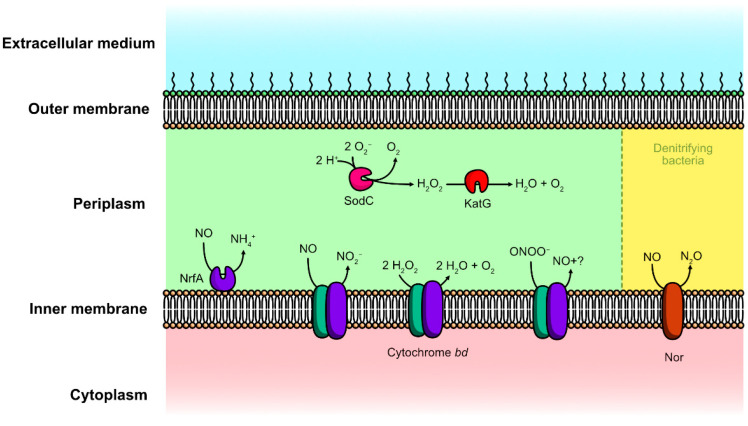
Various mechanisms used by Gram-negative bacteria to detoxify ROS and RNS in the periplasmic space.

**Figure 3 microorganisms-10-00924-f003:**
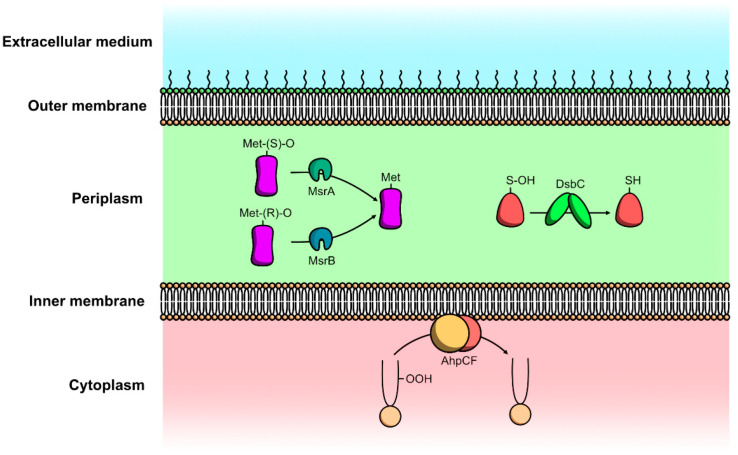
Envelope repair systems of Gram-negative bacteria against oxidative and nitrosative stresses induced damage.

## Data Availability

Not applicable.

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
