# Peer review of "Gram-Negative Bacterial Envelope Homeostasis under Oxidative and Nitrosative Stress"

_microorganisms, 2022, doi:10.3390/microorganisms10050924_

Round 1

Reviewer 1 Report

This manuscript is a review analyzing the effect of oxidative and nitrosative stress in Gram-negative bacteria.

Although the review is well organized, it does not provide new insight into the understanding of stress.

The topic is very well known and there are several reviews related to the topic that has already been covered properly.

  1. Hews Claire L., Cho Timothy, Rowley Gary, Raivio Tracy L. Maintaining Integrity Under Stress: Envelope Stress Response Regulation of Pathogenesis in Gram-Negative Bacteria
    Frontiers in Cellular and Infection Microbiology, vol. 9, 2019  
  2. Sebastian Runkel, Hannah C. Wells, Gary Rowley,
    Chapter Three - Living with Stress: A Lesson from the Enteric Pathogen Salmonella enterica,
    Editor(s): Sima Sariaslani, Geoffrey M. Gadd,
    Advances in Applied Microbiology, Academic Press, Volume 83,
    2013, Pages 87-144.
  3. Flint, Annika, Stintzi, Alain, Saraiva, Lígia M. Oxidative and nitrosative stress defences of Helicobacter and Campylobacter species that counteract mammalian immunity. FEMS Microbiology Reviews, 2016.
  4. Ferric C. Fang, Elaine R. Frawley, Timothy Tapscott, Andrés Vázquez-Torres. Bacterial Stress Responses during Host Infection, Cell Host & Microbe, Volume 20, Issue 2, 2016, Pages 133-143.
  5. Isabelle S. Arts, Alexandra Gennaris, Jean-François Collet,
    Reducing systems protecting the bacterial cell envelope from oxidative damage, FEBS Letters, Volume 589, Issue 14, 2015,
    Pages 1559-1568.

Reviewer 2 Report

The review is devoted to the AO properties of the components of the Gram-negative bacterial envelope. Recently, many reviews have appeared on the AO properties of bacterial cells. However, in this review, these properties are considered from an unusual angle. The paper considers the consequences of both oxidative and nitrosative stress; in addition, attention is paid to the intracellular localization of enzymes. These features and a significant amount of information considered make the article interesting and useful.  I would like to note one more positive feature of the work - along with the works of recent years, old initial works often unfairly forgotten are cited.

I have minor remarks on quoting references - Lanes 64, 73, 74, 106.

Reviewer 3 Report

Dear Authors, 

Your article is well and clearly written, however, I could not see any new idea in it, only a listing of already existing, well known facts. 

Therefore, unfortunately, I cannot recommend your manuscript for publication. Hope you will reconsider the concept of your review, make its novelty more obvious and submit it for reviewing again.

Reviewer 4 Report

The authors: Thibault Chautrand, Djouhar Souak, Sylvie Chevalier, Cécile Duclairoir-Poc

have submitted a draft (Manuscript ID: microorganisms-1666220) entitled „ Gram-negative bacterial envelope homeostasis under oxidative and nitrosative stress“ to the section: Environmental Microbiology. This review could become a valuable contribution for the readers of the corresponding section of the journal Microorganisms. However, I strongly recommend to discuss the suggested publications, especially, under considerations of defensins and lysozymes in this context.

Hews CL, Cho T, Rowley G and Raivio TL (2019) Maintaining integrity under stress: Envelope stress response regulation of pathogenesis in gram-negative bacteria. Front. Cell. Infect. Microbiol. 9:313. doi: 10.3389/fcimb.2019.00313

Dam S, Pagès JM, Masi M. Stress responses, outer membrane permeability control and antimicrobial resistance in Enterobacteriaceae. Microbiology (Reading). 2018 Mar;164(3):260-267. doi: 10.1099/mic.0.000613. Epub 2018 Jan 25. PMID: 29458656.

Ragland SA, Criss AK (2017) From bacterial killing to immune modulation: Recent insights into the functions of lysozyme. PLoS Pathog 13(9): e1006512. https://doi.org/10.1371/journal.ppat.1006512

R. Zhang, L. Wu, T. Eckert, M. Burg-Roderfeld, M. A. Rojas-Macias, T. Lütteke, V. B. Krylov, D. A. Argunov, A. Datta, P. Markart, A. Günther, B. Nordén, R. Schauer, A. Bhunia, M. A. Enani, M. Billeter, A. J. Scheidig, N. E. Nifantiev, H.-C. Siebert (2017) Lysozyme’s lectin-like characteristics facilitates its immune defense function. Q. Rev. Biophys. 50, e9, 1-12.

Lehrer RI, Jung G, Ruchala P, Andre S, Gabius HJ, Lu W. Multivalent binding of carbohydrates by the human alpha-defensin, HD5. J Immunol. 2009 Jul 1;183(1):480-90. doi: 10.4049/jimmunol.0900244. PMID: 19542459.

Heesterbeek DAC, Muts RM, van Hensbergen VP, de Saint Aulaire P, Wennekes T, Bardoel BW, et al. (2021) Outer membrane permeabilization by the membrane attack complex sensitizes Gram-negative bacteria to antimicrobial proteins in serum and phagocytes. PLoS Pathog 17(1): e1009227. https://doi.org/10.1371/journal.ppat.1009227

Round 2

Reviewer 1 Report

The authors have submitted a new version of the manuscript. There is limited information about ROS and RNS damage to the cell envelope. Instead, the authors have included only a very small paragraph to develop this topic. Since the focus of the review is on the envelope, there is limited information for the reader.

Reviewer 3 Report

Dear Authors,

I'm still very skeptical about your review, but now it makes a much better impression than the first time. I hope it will be in demand among the audience.

Please check the Figure 1 carefully, I suspect that there is an error in the direction of the arrow near the upper left proton.
